# Effects of Isocaloric Fructose Restriction on Ceramide Levels in Children with Obesity and Cardiometabolic Risk: Relation to Hepatic De Novo Lipogenesis and Insulin Sensitivity

**DOI:** 10.3390/nu14071432

**Published:** 2022-03-30

**Authors:** Emily Olson, Jung H. Suh, Jean-Marc Schwarz, Susan M. Noworolski, Grace M. Jones, John R. Barber, Ayca Erkin-Cakmak, Kathleen Mulligan, Robert H. Lustig, Michele Mietus-Snyder

**Affiliations:** 1Department of Pediatrics, George Washington University, Children’s National Hospital, Washington, DC 20010, USA; eolson@childrensnational.org; 2Children’s Hospital Oakland Research Institute, University of California Benioff Children’s Hospital Oakland, Oakland, CA 94609, USA; drjungsuh@gmail.com; 3Department of Basic Sciences, Touro University California, Vallejo, CA 94592, USA; jschwarz@touro.edu (J.-M.S.); gjones7@touro.edu (G.M.J.); 4Department of Radiology, University of California San Francisco, San Francisco, CA 94143, USA; susan.noworolski@ucsf.edu; 5Clinical and Translational Science Institute, Children’s National Hospital, Washington, DC 20010, USA; jrbarber@childrensnational.org; 6Department of Pediatrics, Division of Endocrinology, University of California San Francisco Benioff Children’s Hospital, San Francisco, CA 94143, USA; ayca.erkin-cakmak@ucsf.edu; 7Department of Medicine, Division of Endocrinology and Metabolism, University of California San Francisco, San Francisco, CA 94110, USA; kathleen.mulligan@ucsf.edu; 8Department of Pediatrics, University of California San Francisco, San Francisco, CA 94143, USA; robert.lustig@ucsf.edu; 9Department of Pediatrics, Children’s National Hospital, Division of Cardiology, George Washington University, Washington, DC 20010, USA

**Keywords:** sphingolipid ceramide, cardiometabolic risk, insulin sensitivity, childhood obesity

## Abstract

Sugar intake, particularly fructose, is implicated as a factor contributing to insulin resistance via hepatic de novo lipogenesis (DNL). A nine-day fructose reduction trial, controlling for other dietary factors and weight, in children with obesity and metabolic syndrome, decreased DNL and mitigated cardiometabolic risk (CMR) biomarkers. Ceramides are bioactive sphingolipids whose dysregulated metabolism contribute to lipotoxicity, insulin resistance, and CMR. We evaluated the effect of fructose reduction on ceramides and correlations between changes observed and changes in traditional CMR biomarkers in this cohort. Analyses were completed on data from 43 participants. Mean weight decreased (−0.9 ± 1.1 kg). The majority of total and subspecies ceramide levels also decreased significantly, including dihydroceramides, deoxyceramides and ceramide-1-phoshates. Change in each primary ceramide species correlated negatively with composite insulin sensitivity index (CISI). Change in deoxyceramides positively correlated with change in DNL. These results suggest that ceramides decrease in response to dietary fructose restriction, negatively correlate with insulin sensitivity, and may represent an intermediary link between hepatic DNL, insulin resistance, and CMR.

## 1. Introduction

Metabolic syndrome and prediabetes are increasing in prevalence in the United States, now found in 19–35% of youth and up to 54% in adults older than 60 years of age [1]. Cardiometabolic risk (CMR) factor clustering shows stability from childhood into adulthood, with stronger inter-age correlations of risk factors for persons at higher risk [2]. Between 2001 and 2017, the number of youth aged 10–19 with type 2 diabetes (T2DM) increased from 34 to 67 per 100,000 [3]. It follows that cardiovascular mortality is rising fastest for middle-aged men and women in the United States [4], an outcome of the rising prevalence of child obesity and associated cardiometabolic risk (CMR) over the past 40 years. Early identification and intervention to reverse CMR in youth is paramount.

Ceramides are bioactive sphingolipids formed from the combination of a sphinganine long chain base coupled with variable length fatty acid moieties. Ceramides are ubiquitous in cell membranes with both structural and signaling functions, and they serve as coordinators of cell differentiation, proliferation, inflammation, and apoptosis. Ceramides may dysregulate mitochondrial function and contribute to the lipotoxicity of insulin resistance [5]. Insulin resistance has been identified as a central common pathway resulting in dyslipidemia, notably (increased triglycerides (TG) and decreased high-density lipoprotein cholesterol (HDL-c)), central adiposity, hyperglycemia, and hypertension. Taken together, these traditional CMR factors are emblematic of the metabolic syndrome that is associated with a two-fold increase in cardiovascular disease (CVD) [6].

The Western diet, and particularly sugar, has been implicated as a factor contributing to the biochemical changes seen in metabolic syndrome [7,8]. Fructose is notably high in sugar sweetened beverages (SSBs) and other ultraprocessed foods, which now account for 67% of the sugar in children’s diets [9]. Fructose possesses several properties that uniquely contribute to CMR [10]. In particular, SSBs were identified as a leading factor associated with premature cardiometabolic mortality [11].

Fructose is metabolized by the liver, where it is a preferred substrate for de novo lipogenesis (DNL). Fructose also promotes reactive oxygen species formation, leading to endoplasmic reticulum (ER) stress [12]; of which ceramide synthesis is a potential result. Elevated serum total ceramides, dihydro-Cers, and deoxy-Cers have been found to both correlate with and predict CVD and T2DM disease severity in adults [13,14,15]. It is not known how early in life these novel sphingolipid biomarkers can signal departure from a healthy metabolism, or whether the trajectory is modifiable.

Our group previously demonstrated that fructose reduction for nine days in children with obesity and metabolic syndrome, keeping other dietary factors constant, decreased DNL and mitigated traditional biomarkers of CMR [16,17,18]. To further elucidate the putative role of dietary sugar and CMR in this post hoc analysis, we analyzed pre- and post-fructose restriction plasma aliquots for ceramide levels and correlated them with changes in DNL and insulin sensitivity.

## 2. Materials and Methods

Sphingolipid ceramide isolation was performed on samples collected at baseline and following nine days of a pediatric isocaloric fructose reduction dietary trial. Details are outlined in the original study as approved by the UCSF Committee on Human Research and the Touro University Institutional Review Board [16]. An abbreviation of the methods is provided here. Recruitment was restricted to Latino and African American youth at increased CMR who self-identified as high sugar consumers. Eligibility criteria included (1) ages 8–18 years; (2) obesity; (3) at least one additional comorbidity (hypertension, hypertriglyceridemia, impaired fasting glucose, hyperinsulinemia, elevated ALT, or severe acanthosis nigricans). Exclusion criteria included known diabetes, steroid use, medication use that affects insulin secretion or resistance, alcohol use, pregnancy, or neuroactive medications.

On Day 0, participants had their anthropometric measurements recorded, bloodwork, oral glucose tolerance testing (OGTT), magnetic resonance imaging scan (MRI), and dual-energy x-ray absorptiometry (DXA) scan completed. They were provided with nine days of food in three separate installments and instructions to report weight daily. The diet was planned to keep calories consistent with their baseline diet, but total dietary sugar was reduced from 28% to 10% of total calories, and fructose was reduced from 12% to 4% of total calories, respectively. If weight loss was noted from the daily weight reports, then additional food items were provided to maintain weight stability. On Day 10, participants returned to repeat all studies completed on Day 0.

Sphingolipidomic analyses were performed on reserve EDTA plasma stored at −70 °C. Ultra-performance liquid chromatography-tandem mass spectrometry (UPLC-MS/MS, Agilent Technologies, Santa Clara, CA, USA) by JS was used to complete the sphingolipidomics, using validated techniques to quantify sphingolipid ceramides. For these analyses, plasma samples (100 μL) were spiked with 10 μL of internal standard mix and subsequently extracted based on protocols described by Bielawski et al. [19] Ceramide/Sphingoid Internal Standard Mixture I (Avanti polar lipids; LM6002, Avanti Polar Lipids, Birmingham, AL, USA) was used as an internal standard mix. The detection system was composed of an Agilent 1290 (Agilent Technologies, Santa Clara, CA, USA) binary gradient ultra-high pressure chromatography system coupled with an Agilent 6490 (Agilent Technologies, Santa Clara, CA, USA) triple quadrupole mass spectrometer. Sphingolipid metabolites were resolved on a Zorbax RRHD Eclipse Plus C18 column (2.1 × 50 mm; 1.8 micron; Agilent Technologies, Santa Clara, CA, USA) fitted with a pre-column composed of identical matrix. Samples were eluted from the column using a binary gradient composed of mobile phase A (2 mM ammonium formate (Sigma Aldrich, St. Louis, MO, USA) and 0.2% formic acid (Sigma Aldrich, St. Louis, MO, USA) in 18 mΩ water) and mobile phase B (1 mM ammonium formate and 0.2% formic acid in MS-grade methanol (Thermo Fisher Scientific, Waltham, MA, USA) at a flow rate of 1 mL/min. Initial composition was 75% B, and this was increased to 80%, 85%, and 100% at 3, 3.1, and 10 min, respectively. The gradient was maintained at 100% B until 8.5 min and subsequently changed back to initial conditions until the end of the run at 10 min. Analysis was carried out using a multiple-reaction-monitoring mode. For all compounds, the general source settings in the positive ionization modes were as follows: gas temperature 200 °C; gas flow, 14 L min^−1^; nebulizer 20 psi; sheath gas temperature 250 °C; sheath gas flow 11 L min^−1^; capillary voltage 3000 V; and nozzle voltage 0 V. The fragmentor voltage of 380 V and a dwell time of 15 ms were used for all mass transitions, and both Q1 and Q3 resolutions were set to nominal mass unit resolution.

The subset of traditional biomarkers of cardiometabolic risk used in this post-hoc analysis are hemoglobin A1c (HbA1c), weight, systolic blood pressure (SBP), diastolic blood pressure (DBP), alanine aminotransferase (ALT), uric acid (UA), fat mass quantified by DXA scan, both visceral adipose tissue (VAT) and subcutaneous adipose tissue (SAT) quantified by magnetic resonance imaging (MRI), DNL area under the curve (DNL-AUC) by an 8 hr stable isotope feeding assay [17], peak glucose after OGTT (OGTT-peak GLU), peak insulin after OGTT (OGTT-peak INS), hepatic fat fraction quantified by magnetic resonance spectroscopy (HFF), homeostatic model assessment for insulin resistance (HOMA-IR), triglyceride level (TG), low-density lipoprotein cholesterol (LDL-c), high-density lipoprotein cholesterol (HDL-c), free fatty acids (FFA), and composite insulin sensitivity index (CISI) [20].

Paired t-tests were used to compare pre-post plasma ceramide levels between Day 0 and Day 10. Correlation analyses were completed between the changes in plasma sphingolipid ceramides and changes previously reported in the above identified subset of traditional biomarkers [16,17,18]. Results with *p* < 0.05 were considered significant. No correction for multiplicity was made, given the exploratory nature of this research, the limited sample size, and the very strong correlations among the many different biochemically interrelated sphingolipid ceramide subtypes that would have resulted in too conservative of a correction. All data used in these analyses are in the Appendix A.

## 3. Results

Clinical and anthropometric parameters are reported in the original reports [16,17], and a summary of results relevant to these analyses are summarized here. There were 43 pairs of baseline (D0) and post-intervention data (D10) available for sphingolipid ceramide analyzes. The mean age of participants was 13.3 ± 2.7 years. As previously reported, fat mass (by DXA) did not change significantly, but MRI data documented significant reductions in VAT and hepatic fat (HFF), and there was a significant reduction in DNL-AUC. SBP did not change, but DBP decreased. There was improved insulin sensitivity, associated with a significant decrease in each for fasting glucose, glucose area under the curve, fasting insulin, HOMA-IR, OGTT-peak INS, insulin AUC, and a significant increase in CISI over the 10-day period. TG, LDL-c, HDL-c, and AST each declined significantly, while ALT declined non-significantly. Serum free fatty acids increased, consistent with increased peripheral lipolysis and availability of fatty acids for beta-oxidation.

Table 1 lists the changes in total concentration of the four primary ceramide subtypes, and multiple ceramide metabolites within each subtype, for Day 0 and Day 10. Figure 1 shows box plots for the changes in all ceramide species. Three ceramide categories decreased significantly over the ten-day trial: Total Ceramide (*p* = 0.037), Total Cer-1-P (*p* = 0.021), and Total deoxy-Cer (*p* = 0.008). Total dhCer trended downward (*p* = 0.083). Specific ceramide and dhCer subtypes (Cers C24:0, C22:0, C20:0 and C14:0; dhCers C26:0, C22:0 and C20:0) decreased and all five subtypes of deoxy-Cer measured (deoxy-Cers C24:0, C22:0, C20:0, C18:0, C16:0) decreased significantly.

Relationships between change in CMR biomarkers already described in this trial [16,17] and the change in Cers are reported in Table 2 and represented in the correlogram of Figure 2 for the biomarkers with significant correlations. Change in the total level for each of the four primary ceramide species evaluated showed a significant inverse relationship with insulin sensitivity, as measured by CISI. Change in CISI had a significant inverse correlation with changes in 18 out of the 25 ceramide levels included in these analyses, including all deoxy-Cer levels. Furthermore, trends toward inverse correlations emerged with Cer C18:0, dhCer C18:0, dhCer C16:0, dhCer C20:0, and dh Cer C26:0. An inverse correlation was also observed between the change in FFA and change in ceramides, mostly deoxy-Cer levels (Total and C24:0, C22:0, C18:0 and C16:0 deoxy-Cers as well as ceramide C 18:1).

There was a small but significant decrease in weight (−0.9 ± 1.1 kg, *p* < 0.001) over this short trial. Sensitivity analysis of the 33 participants who lost weight (−1.4 + 0.77 kg, *p* < 0.001) and the 10 participants who gained weight (+0.6 ± 0.5 kg, *p* = 0.002) showed comparable correlations between total Cer subtypes and CISI (data not shown). A significant reduction in HFF (hepatic fat fraction) has been described for this study cohort [17]. Only one positive correlation emerged between change in HFF and deoxy-Cer C20:0 (r = 0.34, *p* = 0.035), but change in total deoxy-Cers showed a significant positive correlation with change in DNL-AUC (Table 2, Figure 2). In addition, the following deoxy-Cer subtypes showed significant positive correlation with DNL-AUC: deoxy-Cers C22:0 (r = 0.42, *p* = 0.007), C18:0 (r = 0.41, *p* = 0.008), and C16:0 (r = 0.39, *p* = 0.012).

## 4. Discussion

Conservative estimates using traditional cardiometabolic risk (CMR) biomarkers suggest that only 25% of American adolescents have optimal metabolic health [21]. Thus, up to 75% are at heightened risk for cardiometabolic complications including heart disease, which has been the leading cause of death in the United States for over a century, even during the COVID-19 pandemic [22]. CVD is now also the leading cause of global mortality [23]. Furthermore, two diseases, nonalcoholic fatty liver disease [24] and T2DM [25], rarely seen in children before this century, are now highly prevalent [26]. Both of these diseases are associated with high fructose consumption, SSBs, and ultraprocessed food [27]. A 25 × 25 Global Action Plan, aiming for a 25% reduction in premature mortality from noncommunicable diseases, encompasses nine targets focused on primary and primordial prevention with heart-healthy, evidence-based behavioral changes and improved access to health care resources [28]. However, avoidance of added fructose and fructose-containing sugars, as distinguished from naturally sourced fructose within fiber-rich whole fruit, has not been emphasized as an important lifestyle recommendation. The relationship between fructose and hepatic inflammation through activation of *c-jun* N-terminal kinase is well documented [29]. Our findings about ceramides provide yet another mechanistic link between fructose consumption, DNL, and inflammation. Perhaps the identification of specific biomarkers of dysmetabolism can help direct critical health care resources across lifespans towards those at heightened risk. Ceramides may serve such a purpose.

Ceramides are synthesized de novo from condensation of saturated fatty acid (SFA), specifically palmitoyl-CoA, and amino acid, typically serine. Dietary SFAs are routinely considered to be the environmental trigger for elevated plasma ceramide levels. One adult nutritional intervention showed that dietary SFA was more potent than dietary sugar in increasing circulating ceramide levels [30]. However, the initial major product of DNL is palmitic acid (C16:0) [31], derived from dietary sugar consumption [32]. The Framingham Offspring Study identified a positive association between SSBs and Cers C16:0, C22:0, and C24:0 levels [33]. We have assessed the role dietary sugar reduction in reducing DNL and ceramide levels in a nine-day pediatric nutritional intervention. We noted decreased DNL and improved glucose tolerance, insulin sensitivity, lipid profile, and significant decreases in multiple plasma ceramide metabolites, notably Cers C14:0, C20:0, 22:0, and 24:0. Furthermore, we observed improved insulin sensitivity correlated with reduction in total ceramides and multiple ceramide metabolites (Cers C14:0, 22:0 and 24:0). Because dihydro Cers are precursors to ceramides via the de novo ceramide synthetic pathway, and Cers-1-P are important metabolic ceramide byproducts that mediate inflammatory responses [34], it follows that comparable inverse relationships with insulin sensitivity emerged between total and several specific dhCers measured, as well as with total Cer-1-P and Cers-1-P C16:0 and C18:1. These findings are consistent with the growing body of evidence identifying ceramides as contributing factors to lipotoxicity, insulin resistance, and associated CMR [35].

In animal models, inhibition of ceramide synthesis is both insulin sensitizing and cardioprotective [36,37,38]. Lower ceramide levels may improve management of adult CMR [39,40], and pharmacologic interventions to reduce ceramide levels for effective CMR reduction have been postulated [5]. The question of nutritional modulation to improve ceramide levels has also been addressed. The PREDIMED study showed that the Mediterranean diet improved CVD outcomes in adults with high baseline plasma ceramide levels, although levels did not change [41]. In another study, the Nordic Diet, high in fiber-rich whole grains, fruits and vegetables, and cold-water fish, lowered ceramide levels but insulin regulation was not evaluated [42]. In this study, we showed that a targeted fructose restriction reduced ceramide levels and improved insulin sensitivity.

The changes observed in these three interrelated metabolites, ceramides, dhCers, and Cers-1-P, were not associated with a change in dietary fat, but rather an isocaloric reduction in dietary sugar from 28% to 10% of total calories and in dietary fructose from 10% to 4% of total calories. This suggests that dietary simple sugar, notably dietary fructose, may also play a role in ceramide synthesis. Ceramide could be generated from newly formed palmitate, which would be consistent with their reduction coincident with reduced DNL. Dietary fat was held constant in the meals provided, and peripheral lipolysis of adipose tissue also seems an unlikely source of change in ceramides that moved in an opposite direction to FFA. Our study cannot, however, discern the fat pool used to make ceramides. 

In an adult three-week study of 1000 daily excess calories as saturated fat, unsaturated fat, or carbohydrate, weight predictably increased, but while both excess saturated fat and excess simple carbohydrate triggered DNL, only excess saturated fat was associated with an increase in ceramides C24:0 and C24:1, as well as dhCer C24:0 [30]. These three ceramides did not decrease significantly in our fructose reduction trial, although the change in all three did inversely correlate with change in insulin sensitivity. Change correlations were not reported in the Finnish adult study.

A unique finding in this study was the presence at baseline of deoxy-Cer at levels comparable to circulating plasma ceramides. Deoxy-Cer is generated by a non-canonical or atypical de novo synthetic pathway in which alanine or (less frequently) glycine are condensed with palmitoyl-CoA in lieu of serine [43]. Since neither alanine nor glycine has the hydroxyl group conferred by serine, the resultant deoxy-Cer lacks a hydroxyl group needed for further sphingolipid metabolism, and its toxicity is attributed at least in part to this metabolic full stop that precludes further degradation. In a double-blinded lipidomics screen of plasma from adult patients with normal, steatotic, or fibrotic liver disease, deoxy-dhCers were among the plasma metabolites that distinguished the progression from NAFLD to NASH [44]. Furthermore, deoxy-Cers are toxic for insulin-producing pancreatic beta-cells, and have been linked to lipotoxicity in multiple cardiometabolic conditions [45,46]. Total deoxy-Cers and all deoxy-Cer metabolites measured (deoxy-Cers C16:0, C18:0, C20:0, C22:0, and C24:0) decreased significantly after nine days of isocaloric fructose reduction. As observed for canonical ceramides, the change correlated inversely with improved insulin sensitivity. Interestingly, with the exception of dhCer C18:0, deoxy-Cers were the only metabolites in these analyses with reductions over the short fructose-reduction trial showing a direct association with the measured reduction in DNL, though not with change in the actual hepatic fat fraction.

There was a positive association between the changes observed in almost all ceramide metabolites assayed and the favorable reduction in LDL-c previously described in this study [16,18]. This is predictable because plasma sphingolipid ceramides are carried in lipoproteins. [47]. Study of the potential mechanistic implications of ceramide content for lipoprotein levels and function is evolving rapidly and supports a significant independent contribution of “hyperceridemia” to cardiometabolic toxicity [48]. Only certain Cer metabolites in these analyses were also associated with plasma TG, notably dhCer C26:0, as well as Cer C14:0, Cer-1-P C16:0, and deoxy-Cers C20:0 and C22:0, implicating TG metabolism with flux in sphingolipid pathways in youth with obesity and insulin resistance.

The small changes observed in weight in this short study, despite every effort to keep weight stable, showed a positive but mostly nonsignificant relationship with change in ceramide metabolites (with the exception of deoxy-Cer C18:0, Table 2). It has been demonstrated by MRI that the first weight lost in adults with obesity after lifestyle change is ectopic adiposity, and loss of hepatic lipid is the strongest predictor of improved insulin sensitivity [49]. A 22% reduction in hepatic fat after nine days of isocaloric fructose restriction was measured in our cohort [17], which may be the primary driver of ceramide reduction.

There are several limitations to this study. There was no external control group; on the other hand, five separate internal controls validated our results [16], and our original results have been corroborated by other investigators [50,51]. The sample size is relatively small and the dietary intervention was only nine days in duration. There was a wide variation in individual responses, as illustrated by the broad range of percent change in ceramides shown in Table 1. A study with a larger sample size is needed to better understand predictors of response, to elucidate the relationships between the different CMR biomarkers and sphingolipid ceramides, and to clarify causality between fructose reduction and the results of this study. However, documented significant improvement in several traditional cardiometabolic risk biomarkers, as well as the measured reduction in DNL and hepatic steatosis, make plausible the concomitant changes in ceramide metabolism [17]. In addition, while the study was designed to control weight and isolate fructose reduction as a variable, approximately 75% of the participants still experienced a small amount of weight loss (<1 kg) during the short study period, and it is impossible to dissociate the observed favorable reduction in ceramides from change in adiposity. Indeed, the association between reduced deoxy-Cers and DNL suggests that clearance of ectopic and lipotoxic fat stores upon dietary fructose reduction may mediate both changes.

## 5. Conclusions

This study suggests that dietary fructose stimulates both canonical and atypical ceramide synthetic pathways, possibly by triggering DNL. It also suggests a mechanistic link between fructose and inflammation, which appears to be a major driver of CMR. The significant reduction in toxic ceramide, dhCer, Cer-1-P, and deoxy-Cer metabolites after only nine days of dietary fructose restriction suggests swift improvement in metabolism is possible. Future studies are needed to confirm these findings and to establish whether successful implementation of evidence-based, heart-healthy dietary guidelines that limit added sugars can promote reduction in long-term CMR in at-risk youth.

## Figures and Tables

**Figure 1 nutrients-14-01432-f001:**
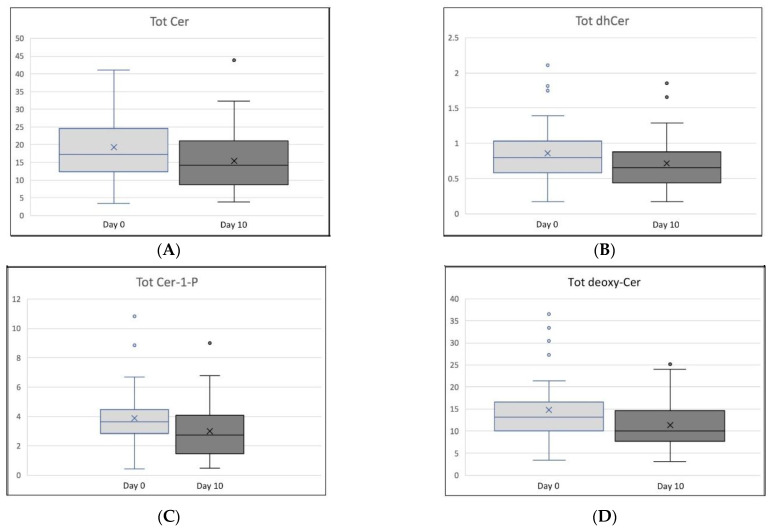
Ceramide Level Box Plots, Day 0 and Day 10. (**A**) Tot Cer = Total Ceramide. (**B**) Tot dhCer = Total dihydro-Ceramide. (**C**) Tot Cer-1-P = Total Ceramide-1-Phospate. (**D**) Tot deoxy-Cer = Total deoxy-Ceramide. Crosses depict means. Hollow circles show outliers.

**Figure 2 nutrients-14-01432-f002:**
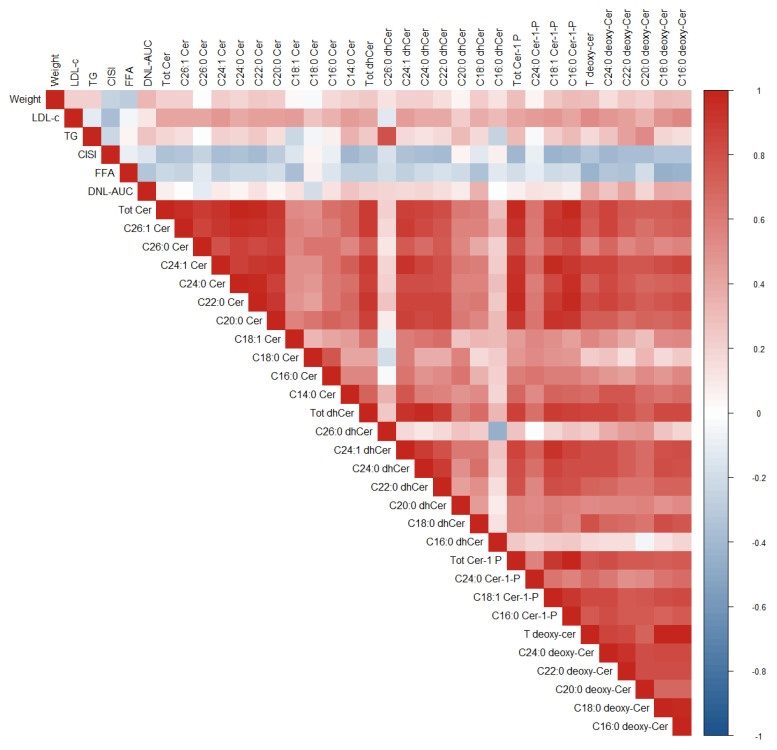
Correlogram of Ceramide, dhCer, Cer-1-P, and deoxy-Cer sphingolipid subspecies versus Weight, LDL-C, TG, CISI, FFA, DNL-AUC for the changes from Day 0 to Day 10. Tot Cer = Total Ceramide. Tot dhCer = Total dihydro-Ceramide. Tot Cer-1-P = Total Ceramide-1-Phospate. Tot deoxy-Cer = Total deoxy-Ceramide. LDL-C = low density lipoprotein cholesterol, TG = triglycerides, CISI = composite insulin sensitivity index, FFA = free fatty acids, DNL-AUC = de novo lipogenesis area under the curve.

**Table 1 nutrients-14-01432-t001:** Ceramide levels by totals and subtypes (*n* = 43).

Ceramide Subtype	Day 0 (Mean ± SD)	Day 10 (Mean ± SD)	Paired *t*-Test (*p*-Value)	% Change (Mean ± SD)
**Tot Cer**	**19.38 ± 8.81**	**15.48 ± 8.44**	**0.037**	**−5.22 ± 114.23**
C26:1 Cer	0.10 ± 0.06	0.09 ± 0.06	0.185	−28.74 ± 187.32
C26:0 Cer	0.22 ± 0.13	0.18 ± 0.11	0.162	−17.80 ± 113.84
C24:1 Cer	2.91 ± 1.43	2.62 ± 1.15	0.240	−34.24 ± 230.44
**C24:0 Cer**	**12.04 ± 6.13**	**9.12 ± 5.89**	**0.028**	**−38.68 ± 311.67**
C22:0 Cer	1.83 ± 0.83	1.42 ± 0.82	0.015	−10.46 ± 179.30
C20:0 Cer	0.84 ± 0.35	0.70 ± 0.34	0.046	−4.9 ± 55.08
C18:1 Cer	0.05 ± 0.03	0.05 ± 0.02	0.999	−33.33 ± 106.17
C18:0 Cer	0.23 ± 0.10	0.23 ±0.09	0.709	+13.48 ± 50.35
C16:0 Cer	0.97 ± 0.32	0.92 ± 0.29	0.391	−1.16 ± 36.29
**C14:0 Cer**	**0.19 ± 0.09**	**0.15 ± 0.07**	**0.005**	**−8.06 ± 67.72**
Tot dhCer	0.86 ± 0.43	0.72 ± 0.38	0.083	−10.34 ± 115.78
**C26:0 dhCer**	**0.08 ± 0.10**	**0.04 ± 0.04**	**0.018**	**−10.00 ± 75.86**
C24:1 dhCer	0.15 ± 0.08	0.14 ± 0.07	0.624	−34.47 ± 169.86
C24:0 dhCer	0.35 ± 0.22	0.31 ± 0.17	0.317	−186.43 ± 991.81
**C22:0 dhCer**	**0.15 ± 0.09**	**0.11 ± 0.06**	**0.021**	**−29.41 ± 292.60**
**C20:0 dhCer**	**0.04 ± 0.02**	**0.03 ± 0.02**	**0.010**	**−9.38 ± 52.34**
C18:0 dhCer	0.03 ± 0.02	0.02 ± 0.01	0.266	−43.72 ± 164.17
C16:0 dhCer	0.06 ± 0.08	0.06 ± 0.12	0.724	−68.07 ± 260.39
**Tot Cer-1-P**	**3.90 ± 1.93**	**2.98 ± 1.88**	**0.021**	**−16.99 ± 211.94**
C24:0 Cer-1-P	0.05 ± 0.02	0.05 ± 0.02	0.673	−13.01 ± 77.38
C18:1 Cer-1-P	0.14 ± 0.07	0.12 ± 0.06	0.122	−45.74 ± 332.11
**C16:0 Cer-1-P**	**3.71 ± 1.86**	**2.82 ± 1.81**	**0.020**	**−19.77 ± 228.53**
**Tot deoxy-Cer**	**14.73 ± 7.50**	**11.35 ± 5.30**	**0.008**	**−0.86 ± 112.62**
**C24:0 deoxy-Cer**	**0.31 ± 0.19**	**0.21 ± 0.16**	**0.003**	**−24.48 ± 290.53**
**C22:0 deoxy-Cer**	**0.24 ± 0.14**	**0.17 ± 0.12**	**0.001**	**−99.83 ± 0.12**
**C20:0 deoxy-Cer**	**0.04 ± 0.03**	**0.03 ± 0.02**	**0.001**	**−15.68 ± 70.41**
**C18:0 deoxy-Cer**	**3.69 ± 2.14**	**2.65 ± 1.36**	**0.003**	**−2.55 ± 116.93**
**C16:0 deoxy-Cer**	**10.45 ± 5.12**	**8.29 ± 3.74**	**0.015**	**−2.77 ± 112.03**

Significant changes are in bold (*p* < 0.05). Tot Cer,= Total Ceramide. Tot dhCer = Total dihydro-Ceramide. Tot Cer-1-P = Total Ceramide-1-Phospate. Tot deoxy-Cer = Total deoxy-Ceramide. C14:0 through C26:0 refer to the length of fully saturated fatty acyl carbon chains linked to the sphingoid base of the ceramide category. C18:1, C24:1 and C26:1 indicate the presence of one unsaturated double bond in the fatty acyl chain. Significant pre-post changes at *p* < 0.05 are indicated by bold font.

**Table 2 nutrients-14-01432-t002:** Pearson Correlation Coefficient, change in plasma ceramide levels with change in each Weight, LDL-C, TG, CISI, FFA, DNL-AUC from Day 0 to Day 10.

Ceramide Subtype	Weight	LDL-C	TG	CISI	FFA	DNL-AUC
		ρ (Prob > |r| under H0: ρ = 0)				
Tot Cer	+0.20 (0.191)	**+0.43 (0.005)**	+0.19 (0.220)	**−0.35 (** **0.021)**	−0.23 (0.135)	+0.07 (0.663)
C26:1 Cer	+0.20 (0.192)	**+0.41 (0.007)**	+0.14 (0.360)	**−0.33 (** **0.032)**	−0.25 (0.101)	+0.02 (0.897)
C26:0 Cer	−0.02 (0.914)	**+0.41 (0.007)**	≤0.01 (0.979)	−0.24 (0.121)	−0.12 (0.442)	−0.11 (0.500)
C24:1 Cer	+0.23 (0.139)	**+0.50 (0.001)**	+0.21 (0.178)	**−0.38 (** **0.013)**	−0.26 (0.091)	+0.10 (0.529)
C24:0 Cer	+0.19 (0.227)	**+0.39 (0.010)**	+0.18 (0.242)	**−0.34 (** **0.025)**	−0.22 (0.151)	+0.06 (0.729)
C22:0 Cer	+0.26 (0.098)	**+0.45 (0.003)**	+0.25 (0.099)	**−0.40 (** **0.009)**	−0.23 (0.139)	+0.14 (0.398)
C20:0 Cer	+0.23 (0.132)	**+0.45 (0.003)**	+0.15 (0.348)	−0.30 (0.051)	−0.18 (0.241)	+0.05 (0.781)
C18:1 Cer	+0.03 (0.844)	**+0.45 (0.003)**	−0.24 (0.125)	−0.15 (0.325)	**−0.36 (0.017)**	+0.14 (0.388)
C18:0 Cer	−0.04 (0.819)	+0.26 (0.091)	−0.04 (0.796)	+0.04 (0.793)	+0.06 (0.690)	−0.19 (0.237)
C16:0 Cer	+0.17 (0.286)	**+0.35 (0.022)**	+0.07 (0.650)	−0.10 (0.536)	−0.16 (0.306)	+0.13 (0.436)
C14:0 Cer	+0.22 (0.159)	**+0.48 (** **0.001)**	**+0.34 (** **0.024)**	**−0.41 (** **0.006)**	−0.27 (0.080)	+0.27 (0.088)
Tot dhCer	+0.25 (0.108)	**+0.43 (0.004)**	+0.25 (0.112)	**−0.35 (0.020)**	−0.28 (0.071)	+0.22 (0.175)
C26:0 dhCer	+0.14 (0.372)	−0.14 (0.360)	**+0.78 (<0.001)**	−0.18 (0.259)	−0.19 (0.224)	+0.21 (0.201)
C24:1 dhCer	+0.22 (0.158)	**+0.46 (0.002)**	+0.17 (0.271)	**−0.35 (** **0.023)**	−0.27 (0.081)	+0.18 (0.258)
C24:0 dhCer	+0.22 (0.161)	**+0.39 (0.011)**	+0.13 (0.398)	**−0.37 (** **0.015)**	−0.22 (0.16)	+0.19 (0.243)
C22:0 dhCer	+0.16 (0.316)	**+0.40 (0.009)**	+0.16 (0.305)	**−0.40 (** **0.008)**	−0.16 (0.298)	+0.16 (0.320)
C20:0 dhCer	+0.06 (0.726)	+0.24 (0.117)	+0.30 (0.050)	+0.06 (0.720)	−0.20 (0.191)	+0.10 (0.543)
C18:0 dhCer	+0.22 (0.151)	**+0.38 (** **0.012)**	+0.16 (0.296)	−0.13 (0.396)	**−0.36 (0.018)**	+0.38 (0.016)
C16:0 dhCer	+0.14 (0.376)	**+0.32 (0.040)**	−0.26 (0.095)	+0.07 (0.678)	−0.12 (0.441)	−0.02 (0.917)
Tot Cer-1-P	+0.30 (0.052)	**+0.44 (0.004)**	+0.30 (0.050)	**−0.42 (** **0.005)**	−0.19 (0.227)	+0.08 (0.616)
C24:0 Cer-1-P	+0.04 (0.776)	**+0.46 (0.002)**	−0.03 (0.866)	−0.10 (0.540)	−0.11 (0.501)	+0.13 (0.438)
C18:1 Cer-1-P	+0.24 (0.120)	**+0.51 (0.001)**	+0.24 (0.125)	**−0.43 (** **0.004)**	−0.21 (0.177)	+0.12 (0.477)
C16:0 Cer-1-P	+0.30 (0.050)	**+0.43 (0.004)**	**+0.30 (** **0.048)**	**−0.42 (** **0.006)**	−0.19 (0.230)	+0.08 (0.627)
Tot deoxy-Cer	+0.30 (0.054)	**+0.55 (** **<0.001)**	+0.17 (0.285)	**−0.34 (** **0.026)**	**−0.43 (0.004)**	**+0.40 (** **0.010)**
C24:0 deoxy-Cer	+0.14 (0.357)	**+0.48 (0.002)**	+0.27 (0.077)	**−0.42 (** **0.005)**	**−0.34 (0.026)**	+0.28 (0.082)
C22:0 deoxy-Cer	+0.26 (0.094)	**+0.46 (** **0.002)**	**+0.42 (** **0.005)**	**−0.38 (** **0.013)**	**−0.35 (0.023)**	**+0.42 (** **0.007)**
C20:0 deoxy-Cer	+0.21 (0.174)	**+0.40 (0.008)**	**+0.53 (<** **0.001)**	**−0.37 (** **0.016)**	−0.19 (0.213)	+0.20 (0.227)
C18:0 deoxy-Cer	**+0.31 (0.044)**	**+0.49 (** **0.001)**	+0.19 (0.216)	**−0.34 (** **0.027)**	**−0.46 (0.002)**	**+0.41 (** **0.008)**
C16:0 deoxy-Cer	+0.29 (0.058)	**+0.56 (** **<0.001)**	+0.14 (0.365)	**−0.33 (** **0.032)**	**−0.42 (0.005)**	**+0.39 (** **0.012)**

Significant changes are in bold (*p* < 0.05). Tot Cer = Total Ceramide. Tot dhCer = Total dihydro-Ceramide. Tot Cer-1-P = Total Ceramide-1-Phospate. Tot deoxy-Cer = Total deoxy-Ceramide. LDL-C = low density lipoprotein cholesterol, TG = triglycerides, CISI = composite insulin sensitivity index, FFA = free fatty acids, DNL-AUC = de novo lipogenesis area under the curve. Plus signs indicate positive correlations. Minus signs indicate negative correlations. Significant correlations at *p* < 0.05 are in bold font.

## Data Availability

Data is contained within the article or Appendix A. The original data presented in this study are available in Lustig R.H., Mulligan K., Noworolski S.M., et al. Isocaloric fructose restriction and metabolic improvement in children and obesity and metabolic syndrome [16]. The ceramide data are available in the Appendix A.

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
