# Peer review of "Effects of Isocaloric Fructose Restriction on Ceramide Levels in Children with Obesity and Cardiometabolic Risk: Relation to Hepatic De Novo Lipogenesis and Insulin Sensitivity"

_nutrients, 2022, doi:10.3390/nu14071432_

Round 1

Reviewer 1 Report

The authors evaluate the effects of isocaloric fructose restriction on ceramide levels in children with obesity and metabolic syndrome.

Overall, a well organized, well written and nice manuscript, addressing an interesting topic that still needs further investigation and larger sample sizes.

Author Response

We thank reviewer 1 for recognizing that these findings are interesting, and that they invite further prospective investigation with larger sample sizes. We would respectfully add that future research might include expansion of the lipidome to provide further mechanistic insights to the observed associations between isocaloric fructose restriction and ceramide levels in children.

Reviewer 2 Report

The authors performed a trial study to assess the ceramide levels pre- and post-fructose restriction and correlated them with changes in de novo lipogenesis and insulin sensitivity.

The study is original and interesting because fructose intake has been complicated as a factor contributing to insulin resistance.

Major Comments are the recognized limitations of the study. Not having control group and a small sample size. The wide variation in individual responses and the short period of intervention (only 9 days) the few days of intervention make it difficult to draw conclusions from the study

Author Response

We thank reviewer 2 for recognizing that these findings are original and interesting, and agree that there are recognized limitations. We cannot, given the post-hoc nature of these sphingolipid ceramide analyses, change the research design, which was rigorously set up to test the association between fructose and de novo lipogenesis, with each participant serving as their own control.  The study is indeed short, but the changes observed in traditional cardiometabolic risk biomarkers already described make plausible the observation of change in associated but less commonly measured lipidomic biomarkers.  We acknowledge there is wide variation in response, as is common in nutritional interventions. It is impossible to assess potential genetic, epigenetic, and/or compliance roots of this variation with the data available.  While there was significant overall favorable movement in the lipidome, the variation in response strengthens the change correlations enumerated in Table 2 and illustrated in Figure 2.  We agree one cannot draw conclusions from one small study, but believe our observations are hypothesis-generating. Future research will be needed to validate and extend these findings.

Reviewer 3 Report

This manuscript is the research about "Effects of isocaloric fructose restriction on ceramide levels in children with obesity and metabolic syndrome: relation to hepatic de novo lipogenesis and insulin sensitivitly" submitted by Emily Olson , Jung H Suh , Jean-Marc Schwarz , Susan M Noworolski , Grace M Jones , John R Barber , Ayca Erkin-Cakmak , Kathleen Mulligan , Robert H Lustig , Michele Mietus-Snyder * describes the 9-day fructose reduction trial, controlling for other dietary factors and weight, in the children with obesity and metabolic syndrome, decreased DNL and mitigated cardiometablic risk (CMR) biomarkers. The analysis was completed on data from 43 participants. The manuscript writes well, but there are some points that need to be clarified

1.    The author should be providing IRB information
2.    The result of this manuscript describes the “there was a small but significant decrease in weight”, but still 10 participants gained weight, how to prove that this result is consistent with the research inference?

Author Response

We appreciate reviewer 3’s positive overall reaction to our manuscript and have tried to address the points raised for clarification.

  1. Our Institutional Review Board (IRB) approval is mentioned at the start of the Methods section and more detailed information was included following the body of the manuscript, per standard journal requirements, in the Institutional Review Board Statement. We had already included the protocol number approved by the UCSF Human Subjects Committee and have now also added the IRB protocol number provided by Tuoro University California Human Subjects Committee (tracked edit lines 394-396).

  1. We appreciate the reviewer’s question about the change in weight. This issue was integral to the original study design and we specified in the Methods that:

“If weight loss was noted from the daily weight reports, then additional food items were provided to maintain weight stability.”

Despite this effort of course, as described in Results: 

“There was a small but significant decrease in weight (-0.9 + 1.1 kg), p <0.001) over this short trial. Subgroup sensitivity analysis of the 33 participants who lost weight (-1.4 + 0.77 kg, p < 0.001) and the 10 participants who gained weight (+0.6 + 0.5 kg, p = 0.002) showed comparable correlations between total Cer subtypes and CISI (data not shown).

Fluctuation in weight either up or down proved to be small, in part due to the efforts to minimize weight change.  The paradox of a small mean weight loss (for the cohort of 43) while 10 subjects gained a small amount is a testament to the variability of response highlighted by the second reviewer and recognized by our authorship team.  Change in only one deoxy Cer (C18:0) correlated with the change in weight, while change in 3 of the 5 deoxy Cers tested (including C18:0 deoxy Cer) correlated with change in hepatic adiposity. We believe, as included in our Discussion (lines 293-299) that change in adiposity rather than weight per se, is the driver behind these observations, but further study will be necessary to better understand this important issue.